# Measurement properties of pain scoring instruments in farm animals: A systematic review using the COSMIN checklist

**Rubia Mitalli Tomacheuski[1], Beatriz Paglerani Monteiro[2], Marina Cayetano Evangelista[2], Stelio Pacca Loureiro Luna[3], Paulo Vinícius Steagall[1,2,4] ***

1 Department of Anaesthesiology, Medical School (FMB) of São Paulo State University (UNESP), Botucatu, São Paulo, Brazil, 2 Département de sciences cliniques, Faculté de médecine vétérinaire, Université de Montréal, Saint-Hyacinthe, Québec, Canada, 3 Department of Veterinary Surgery and Animal Reproduction, School of Veterinary Medicine and Animal Science, São Paulo State University (UNESP), Botucatu, São Paulo, Brazil, 4 Department of Veterinary Clinical Sciences and Centre for Companion Animal Health and Welfare, Jockey Club College of Veterinary Medicine and Life Sciences, City University of Hong Kong, Hong Kong, China

* paulo.steagall@umontreal.ca

This is a Registered Report and may have an associated publication; please check the article page on the journal site for any related articles.

## Abstract

This systematic review aimed to investigate the measurement properties of pain scoring instruments in farm animals. According to the PRISMA guidelines, a registered report protocol was previously published in this journal. Studies reporting the development and validation of acute and chronic pain scoring instruments based on behavioral and/or facial expressions of farm animals were searched. Data extraction and assessment were performed individually by two investigators using the Consensus-based Standards for the Selection of Health Measurement Instruments (COSMIN) guidelines. Nine categories were assessed: two for scale development (general design requirements and development, and content validity and comprehensibility) and seven for measurement properties (internal consistency, reliability, measurement error, criterion and construct validity, responsiveness and cross-cultural validity). The overall strength of evidence (high, moderate, low, or very low) of each instrument was scored based on methodological quality, number of studies and studies' findings. Twenty instruments for three species (bovine, ovine and swine) were included. There was considerable variability concerning their development and measurement properties. Three behavior-based instruments scored high for strength of evidence: UCAPS (Unesp-Botucatu Unidimensional Composite Pain Scale for assessing postoperative pain in cattle), USAPS (Unesp-Botucatu Sheep Acute Composite Pain Scale) and UPAPS (Unesp-Botucatu Pig Composite Acute Pain Scale). Four instruments scored moderate for strength of evidence: MPSS (Multidimensional Pain Scoring System for bovine), SPFES (Sheep Pain Facial Expression Scale), LGS (Lamb Grimace Scale) and PGS-B (Piglet Grimace Scale-B). Most instruments (n = 13) scored low or very low for final overall evidence. Construct validity was the most reported measurement property followed by criterion validity and reliability. Instruments with reported validation are urgently required for pain assessment of buffalos, goats, camelids and avian species.

**Data Availability Statement:** All relevant data are within the manuscript and its Supporting information files.

**Funding:** FAPESP (2017/12815-0; Recipient Stelio Pacca Loureiro Luna), Coordenacao de Aperfeicoamento de Pessoal de Nivel Superior (CAPES; recipient Rubia Mitalli Tomacheuski), Natural Sciences and Engineering Research Council of Canada (NSERC; RGPIN-2018-03831). The funders had no role in study design, data collection and analysis, decision to publish, or preparation of the manuscript.

**Competing interests:** The authors have declared that no competing interests exist.

# Introduction

Society has been increasingly concerned about the impact of pain on farm animal welfare [1]. Farm animals are less frequently treated for pain when compared with companion animals [2] and horses [3]. Possible reasons for this include the misconception that farm animals do not feel as much pain as small animals, concerns related to withdrawal times of analgesics for human food safety, and lack of knowledge or empathy about pain in farm animal species [3, 4], and budget considerations for the cost of analgesic therapies combined with the low zootechnical and affective value of farm animals [5–8]. Pain causes suffering, fear and stress, negatively impacting animal welfare and sometimes decreasing productivity [5, 9, 10]. Pain recognition and measurement are important components of animal welfare [5].

Pain assessment in animals is commonly performed through evaluation of species-specific behaviors [11] and changes in facial expressions [12–14]. Other methods of pain assessment include the use of quantitative sensory testing for evaluation of the animals' sensory profile [15] and the use of kinetics or kinematics for evaluation of levels of activity and lameness [16–18]. However, these outcome measures require specific equipment and training and are not readily available in practice nor they evaluate the affective and emotional aspects of pain. Surrogate measures of pain might also include animal production outcomes, physiological parameters, and biomarkers [19–21]; yet these are also not necessarily specific to pain. For these reasons, in practice, pain assessment relies on the evaluation of pain-related behaviors (including facial expressions) using pain scoring instruments (i.e. scales, tools, metrology instruments, etc.). Pain scoring instruments are non-invasive, inexpensive, do not require any equipment or restraint and may be performed by remote observation [22]. They are used to identify and quantify pain, and to monitor response to analgesic treatments. These instruments focus on the behavioral expression of pain and generally include a systematic description of behaviors accompanied by their respective scores. When such behaviors only involve facial expressions, they are known by facial expression or grimace scales. Pain scoring instruments have been developed for farm animals and may include assessment of activity, body posture, response to interaction, attention to wound/painful area, and/or facial expressions [14, 22–26]. In ruminants, for example, the most frequently observed pain-related behaviors include changes in appearance, posture, gait, appetite, interaction with other animals and the environment, decreased or increased frequency of locomotion, weight bearing, vocalization, increased attention to the injured area, lip-licking, increased tonus of the lips, teeth grinding, tremors and strong tail wagging [3, 5, 26–28]. Similarly, pain-related behaviors and changes in facial expressions have been identified in swine [14, 22]. In poultry, there is a lack of studies regarding pain assessment; however, change or absence of normal behaviors have been described including decreased social interactions, increased aggression, showing guarding and/or grooming behavior [29]. Unidimensional scales such as the numerical rating scale (NRS), simple descriptive scale (SDS) and visual analog scales (VAS) have been used in the past to measure postoperative pain in sheep [30, 31]. However, these tools are not considered adequate because they were developed and validated for humans who self-report their degree of pain; these scales are subjective, not species-specific and influenced by the level of familiarity/expertise of the observer [26, 32, 33]. Species-specific pain scales have been developed for use in farm animals, such as sheep, cattle and pigs, and different levels of validation have been reported for some of these instruments [14, 22–24, 26, 34, 35]. Nevertheless, there is lack of validated instruments for some species of farm animals, like goats, camels and poultry.

Pain scoring instruments need to undergo several steps of scientific validation to ensure they are valid and reliable before they can be used in practice with confidence. In order to evaluate whether an instrument is valid and reliable, one must assess the measurement (or

psychometric) properties of such instrument. Measurement properties refer to the characteristics or attributes of an instrument which are a consequence of the methodology used in their respective studies. In other words, measurement properties refer to the quality of the methodology. The most commonly reported measurement properties of pain scoring instruments include a) development/content validity (expert assessment of the items included in the scale, the calculation of a content validation index, development of ethogram and/or evidence from the literature [26, 36]), b) structural and/or cross-cultural validity [36–38], c) internal consistency (degree of the interrelatedness among the items [36, 38]), d) measurement error (systematic and random error in a patient's score that is not associated to real changes in the construct to be assessed including sensitivity, specificity and accuracy [38]), e) reliability (whether the scores are consistent between different observers and over time, known as inter- and intra-observer reliability, respectively [22, 36]), f) criterion validity (correlation of the proposed tool with other existent scales [36, 38]), g) construct validity (whether the tool measures what it is supposed to measure by comparing different known groups [36, 38]), h) responsiveness (ability to detect changes over time) and i) a definition of a cut-off point for administration of rescue analgesia [22, 26, 36].

Systematic reviews of outcome measurement instruments (e.g. pain scoring instruments) are important for selecting the most suitable instrument to measure a construct of interest (i.e. pain) in the target study population [39]. To the authors' knowledge, systematic reviews on the evidence of the measurement properties of different pain scoring systems in farm animals have not been published.

## Objective

This systematic review aimed to provide evidence relating to the measurement properties (i.e. reliability, validity and sensitivity) of pain scoring instruments used for pain assessment in farm animals using the Consensus Based Standards for the Selection of Health Measurement Instrument (COSMIN) methodology [38, 40, 41].

## Materials and methods

The study protocol described herein was published before data collection (Registered Report Protocol [42]) according to the PRISMA (Preferred Reporting Items for Systematic Reviews and Meta-Analyses). Reporting of findings were performed according PRISMA and the 10-step COSMIN guidelines.

### Databases and search terms

Five bibliographic databases (MEDLINE via PubMed, EMBASE, Web of Science, and CAB abstracts and Biological Abstracts via Web of Science) were searched to identify studies published in peer-reviewed journals. There was neither publication period nor language restriction. The search terms were defined using MeSH (Medical Subject Headings), a controlled vocabulary thesaurus produced by the National Library of Medicine, which index articles for MEDLINE/PubMed and using DeCS (Health Science Descriptors), a structured and multilingual vocabulary used as a unique language in indexing articles from scientific literature via the Virtual Health Library, which includes databases such as LILACS, MEDLINE, PAHO IRIS Repository, BIGG International database GRADE guidelines, BRISA Regional Base of Health Technology Assessment Reports of the Americas, CARPHA EvIDeNCe Portal, Observatorio Regional de Humanos de Salud, and PIE Evidence-Informed Policies.

The chosen search terms were refined and tested using PubMed. The following descriptor items were included: ("pain scoring system*" OR "pain scale*" OR "pain indicator*" OR

"grimace scale*" OR "facial expression*" OR "pain behavior*" OR "pain assessment*") AND ("farm animal*" OR ruminant* OR bovine OR beef OR cattle OR cow OR cows OR buffalo* OR camel* OR ovine OR sheep* OR lamb* OR goat* OR caprine* OR swine OR porcine OR pig OR pigs OR piglet* OR poultry* OR chicken* OR fowl* OR duck* OR geese).

## Eligibility criteria

Original studies reporting the development and/or validation of pain scoring instruments in farm animals as well as manuscripts reporting the assessment of one or more measurement properties of these instruments, were included. These studies involved naturally-occurring or experimental acute and chronic painful conditions in bovine (beef and dairy cattle, and buffalo), ovine (sheep and lamb), caprine (goat and kid), camel, porcine (pig and piglets) and poultry (chicken, fowl, ducks, turkeys and geese). These species were chosen since they are the most relevant species used for production of animal protein (meat, dairy products and eggs) according to the Organization for Economic Co-operation and Development (OECD) and the Food and Agriculture Organization (FAO) of the United Nations, the OECD-FAO Agricultural Outlook 2020–2029 [43].

Studies that only reported the use of pain scales as an outcome measurement instrument (e.g. in randomized controlled trials comparing two different treatments), studies in which a pain scale was used in the validation of another instrument, studies reporting only ethogram/ list of pain-related behaviors without a scoring system, studies reporting non-ordinal pain assessment variables, or review and systematic reviews were not included. Studies reporting the use of pain scoring instruments to measure constructs other than pain, for example studies assessing animal welfare, in which pain was considered within the overall evaluation, studies assessing nociceptive testing, and studies for which the full text was not available were excluded.

## Literature search

Study titles and their abstracts were screened for eligibility by two investigators (RMT and BPM) using the search strategy described above. Full-text articles were selected, references were exported into Endnote (version X9), Mendeley and Covidence (a web-based software platform integrated with the Cochrane's review production toolkit that streamlines the production of systematic reviews) and duplicates were removed. Full-text articles were independently reviewed for eligibility criteria by two investigators (RMT and BPM) using Covidence. "Snowball" methods such as pursuing references of eligible articles and/or reviews and electronic citation tracking were used to maximize the retrieval of relevant studies.

## Data extraction

Data from included studies were extracted (RMT) using a predefined data collection sheet (Excel file). The following information was extracted: 1—characteristics of the study population (age, gender, breed/strain, where/how animals were housed, how animals were handled, duration and source of pain); 2—characteristics of the scale (name/version, language/translation, scoring method, number and name of items/action units); 3—setting and purpose for which the scale is intended (e.g. chronic or acute pain, adult or juvenile/pediatric animals, hospital, experimental or commercial setting), interpretability and operational characteristics such as the feasibility for users (i.e. time required for completion of the instrument, who the end-users are, whether training is required, whether evaluations could be done in real-time or using image or video assessment).

## Assessment of the measurement properties

The quality assessment and summary of evidence were performed independently by two reviewers (RMT and BPM) using an Excel file. All information were recorded, evaluated systematically and adapted from the COSMIN checklist [38]. The COSMIN aims to improve the selection of outcome measurement instruments in research and clinical practice [38]. Its methodology was specifically developed and validated for use in reviews of patient-reported outcome measures [38, 40, 41, 44]. However, it can be adapted and used for other types of outcome measurement instruments such as those where pain is not self-reported and is evaluated by proxy, which is the case in veterinary medicine. [41]. For these reasons, an adapted COSMIN evaluation sheet was used. Items such as methods of interviewing and comprehensibility (by the patient point of view) were not assessed, although comprehensibility was adapted and assessed with the content validity, on the end-user point of view. The following categories were evaluated: two for scale development (1a. general design requirements and development and 1b. content validity and comprehensibility) and seven for measurement properties (internal consistency, reliability, measurement error, criterion and construct validity, responsiveness and cross-cultural validity). Moreover, interpretability and feasibility were evaluated. If the reviewers (RMT and BPM) were unable to reach a consensus on the assessment of measurement properties, a third reviewer was consulted (MCE).

Each criterion from the nine categories was assessed for methodological quality (Table 1; Part A) and scored as 'very good', 'adequate', 'doubtful', 'inadequate', or 'not applicable'. The lowest score among all criteria for each category was used as the final score for that category [45, 46]. Detailed guidelines used for scoring each criterion are available as supplementary material (S1 Table). Part A was undertaken for each study.

The quality of the findings for each category (Table 2; Part B) was rated as 'sufficient or positive [+]' when the majority of the summarized results met the criteria for good measurement properties, 'insufficient or negative [–]' when the majority of the summarized results did not meet the criteria for good measurement properties, 'conflicting findings [+/-]' or 'indeterminate [?]'. Part B was initially undertaken for each study. Thereafter, all the studies available for each instrument were rated together to produce an overall rating of quality of the findings for each instrument.

The strength of evidence for each category from each instrument was defined (Table 3; Part C) based on the overall methodological quality (Part A) and overall quality of the findings (Part B). The strength of evidence was summarized as 'high', 'moderate', 'low', 'very low' or 'unknown' using a modified Grading of Recommendations, Assessment, Development and Evaluations (GRADE) proposed by the COSMIN guidelines for grading the quality of the evidence in systematic reviews of patient-reported outcome measures [38, 48]. Moreover, the evidence was downgraded in one level (e.g. moderate to low) when there was a serious risk of bias, in two levels (e.g. moderate to very low) if there was a very serious risk of bias, and in three levels (e.g. high to very low) when there was an extremely serious risk of bias [40]. Part C was performed according to a consensus among three investigators (RMT, BPM and MCE). Rating was initially undertaken for each category of each instrument and subsequently used to define an overall strength of evidence for each instrument.

## Results

A total of 864 studies were retrieved, 209 duplicates were removed, 655 studies were screened (title and abstract), and 607 studies were excluded. Finally, 48 full-text studies were assessed for eligibility and 23 were included for data extraction and assessment containing a total of 20 pain scoring instruments (Fig 1).

**Table 1. Criteria used for assessment of methodological quality (Part A).** Adapted from the Consensus-based Standards for the Selection of Health Measurement Instruments (COSMIN) [38, 41, 45, 47].

| Components of the scale | Categories | Criteria |
|---|---|---|
| Scale development | 1a. General design requirements and development | 1. Is a clear description provided of the construct to be measured? |
| | | 2. Is the origin of the construct clear: was a theory, conceptual framework or disease model used or clear rationale provided to define the construct to be measured? |
| | | 3. Is a clear description provided of the target population and context for which the scale was developed? |
| | | 4. Was the scale development study performed in a sample representing the target population?<br>5. Was an appropriate method used to identify relevant items/AU for a new scale?<br>6. Was a skilled observer or group of observers (experts in the field) used to define the items?<br>7. Were the animals undisturbed during evaluation (or was the effect of handling / observer accounted)? |
| | 1b. Content validity and comprehensibility | 1. Was the content validity established? |
| | | 2. Was an appropriate method used to ask professionals whether each item is relevant for the construct of interest? |
| | | 3. Was an appropriate method used to ask professionals whether each item is clear for the construct of interest? |
| | | 4. Does the scale include descriptors of both normal and pain-related behaviors?<br>5. Was the comprehensibility evaluated by the end-user?<br>6. Was an appropriate method used to assess the comprehensibility—regarding to instructions, items, and response options? |
| Measurement properties | 2a. Internal consistency | 1. Was the internal consistency calculated and reported? |
| | | 2. Were there any other important flaws? |
| | 2b. Reliability | 1. Was inter-rater reliability reported? |
| | | 1.1. Was the number of raters appropriate for inter-rater reliability testing? |
| | | 1.2. Was the statistical method for calculating inter-rater reliability appropriate? |
| | | 2. Was intra-rater reliability reported? |
| | | 2.1. Was the time interval appropriate for intra-rater reliability testing? |
| | | 2.2. Were the test conditions similar for the measurements? e.g. type of administration, environment, instructions |
| | | 2.3. Was the statistical method for calculating intra-rater reliability appropriate? |
| | | 3. Were there any other important flaws? |
| | 2c. Measurement error | 1. Were sensitivity, specificity and/or accuracy determined? |
| | | 2. Were there any other important flaws? |
| | 2d. Criterion validity (i.e. comparison with a gold standard or other validated method) | 1. Was criterion validity reported? |
| | | 2. Is it clear what the gold standard or other method measure(s)? |
| | | 3. Were the measurement properties of the gold standard or other validated method adequate? |
| | | 4. Was the statistical method appropriate for the hypotheses to be tested? |
| | | 5. Were there any other important flaws? |
| | 2e. Construct validity (comparison between subgroups—discrimination between painful and pain-free animals) | 1. Was construct validity reported? |
| | | 2. Was an adequate description provided of important characteristics of the subgroups? |
| | | 3. Was the statistical method appropriate for the hypotheses to be tested? |
| | | 4. Were there any other important flaws? |
| | 2f. Responsiveness (discrimination between before and after analgesic intervention) | 1. Was responsiveness reported? |
| | | 2. Was an adequate description provided of the intervention given? |
| | | 3. Was the statistical method appropriate for the hypotheses to be tested? |
| | | 4. Were there any other important flaws? |
| | 2g. Cross-cultural validity | 1. Were translation and back translation performed? |
| | | 2. Were the samples similar for relevant characteristics? |
| | | 3. Were there any other important flaws? |

Each criterion was independently scored by two individuals as 'V' (very good), 'A' (adequate), 'D' (doubtful), 'I' (inadequate) or 'N' (not applicable). AU = action units.

**Table 2. Criteria used for rating the quality of the findings (Part B).** Adapted from the Consensus-based Standards for the Selection of Health Measurement Instruments (COSMIN) [38, 41, 42].

| Components of scale validation | Categories | Rating |
|---|---|---|
| Scale development | **1a. General requirements and development** | (+) The model/stimulus are relevant AND all items refer to relevant aspects of the construct to be measured AND are relevant for target population AND context of use<br>(?) Not all information for (+) reported OR potential biases identified<br>(-) Criteria for (+) not met AND substantial bias identified |
| | **1b. Content validity and comprehensibility** | (+) The items are relevant and both items AND response match AND are clearly worded<br>(?) Not all information for (+) reported OR potential biases identified<br>(-) Criteria for (+) not met AND substantial bias identified |
| Measurement properties | **2a. Internal consistency** | (+) Cronbach's alpha $\geq 0.70$<br>(?) Cronbach's alpha not reported<br>(-) Cronbach's alpha $< 0.70$ |
| | **2b. Reliability** | (+) ICC OR weighted Kappa $\geq 0.70$<br>(?) ICC OR weighted Kappa not reported<br>(-) ICC OR weighted Kappa $< 0.70$ |
| | **2c. Measurement error** | (+) Accuracy $> 80\%$<br>(?) Not defined OR $> 60$ and $< 80\%$<br>(-) Accuracy $< 60\%$ |
| | **2d. Criterion validity (comparison between subgroups—discrimination between painful and pain-free animals)** | (+) Correlations clearly described AND coefficients $\geq 0.70$<br>(?) Correlations not reported<br>(-) Correlations $< 0.70$ |
| | **2e. Construct validity: Comparison between subgroups (discrimination between painful and pain-free animals)** | (+) Results demonstrated a statistically significant difference between groups (discriminant validity/ hypothesis confirmed)<br>(?) No differences between relevant groups reported<br>(-) Results did not demonstrate a difference between groups (hypothesis not confirmed) |
| | **2f. Responsiveness (discrimination between before and after analgesic intervention)** | (+) At least 75% of the results are in accordance with the hypotheses (difference after analgesic intervention)<br>(?) Not reported OR No hypotheses determined<br>(-) Results not in accordance with hypotheses |
| | **2g. Cross-cultural validity** | (+) The translated OR cultural adapted instrument is an adequate reflection of the performance of the items / AU of its original version<br>(?) Not all information for (+) reported OR potential biases identified<br>(-) Criteria for (+) not met AND substantial bias identified. |

ICC: intra-class correlation coefficient. AU: action units. '+' (sufficient or positive; when most of the summarized results meet the criteria for good measurement properties), '-' (insufficient or negative; when the majority of the summarized results do not meet the criteria for good measurement properties), '+/-' (inconsistent/ conflicting findings), or '?' (indeterminate).

**Table 3. Criteria used for summarizing the strength of evidence (Part C).**

| Strength of evidence | Criteria |
|---|---|
| **High** | Consistent findings in multiple studies of at least 'adequate' quality OR one study of 'very good' quality |
| **Moderate** | Conflicting findings in multiple studies of at least 'adequate' quality OR consistent findings in multiple studies of at least 'doubtful' quality OR consistent findings in one study of 'adequate' quality |
| **Low** | Conflicting findings in multiple studies of at least 'doubtful' quality OR one study of 'adequate' quality OR consistent findings in one study of 'doubtful' quality |
| **Very Low** | Only studies of 'inadequate' quality OR conflicting findings in one study of 'doubtful' quality |
| **Unknown** | No studies |

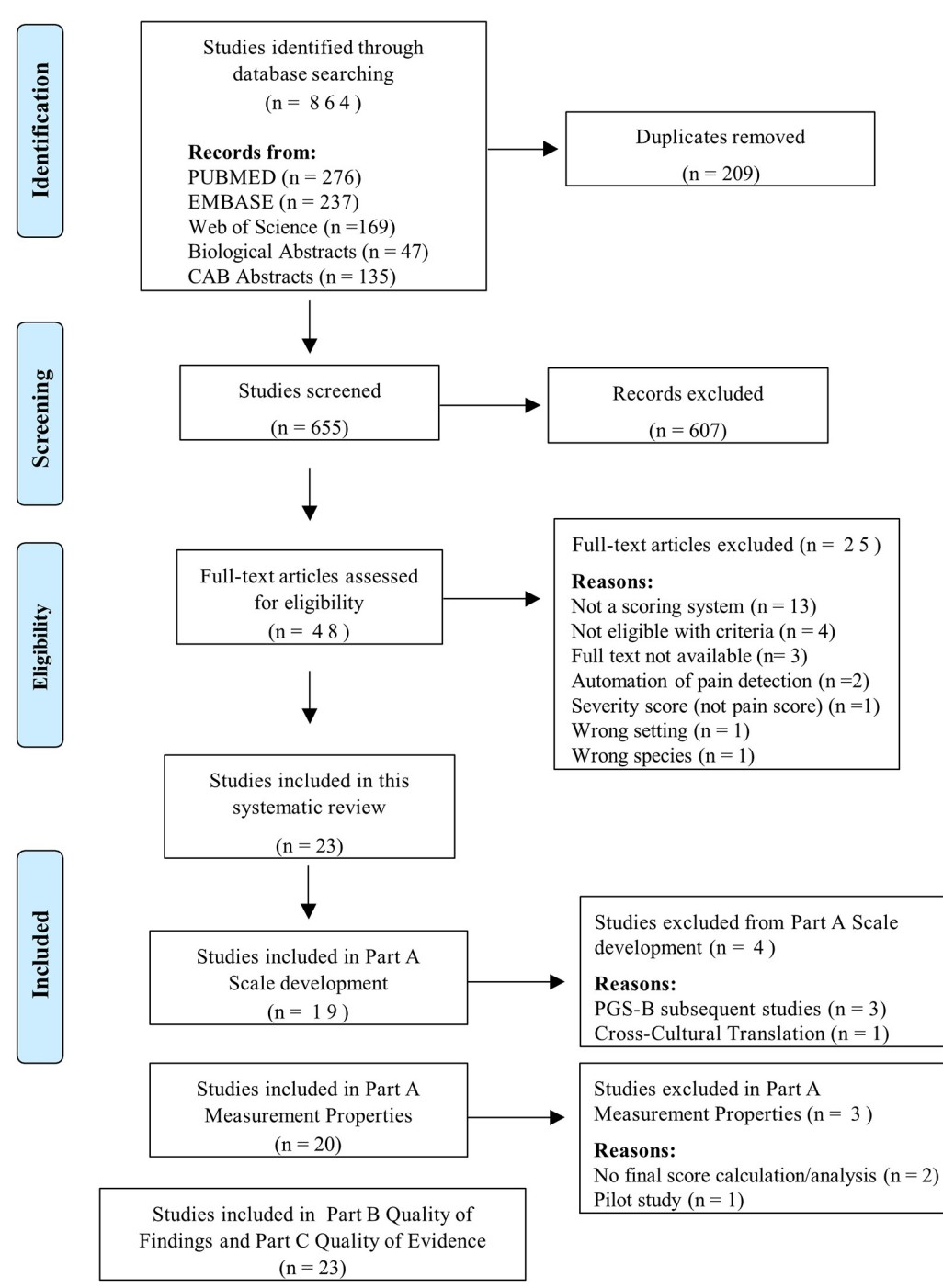

**Fig 1. PRISMA flow diagram of studies on the measurement properties of pain scoring instruments for farm animals.**
*From*: Moher D, Liberati A, Tetzlaff J, Altman DG, The PRISMA Group (2009). *Preferred Reporting Items for Systematic Reviews and Meta- Analyses: The PRISMA Statement. PLoS Med 6(7): e1000097. doi: 10.1371/joumal.pmedl000097 **For more information, visit** www.prisma-statement.org.

A total of 20 pain scoring instruments were included (Table 4). There were 12 behavior-based scales including six for bovine (beef and dairy cattle): *'Unesp-Botucatu Unidimensional Composite Pain Scale for assessing postoperative pain in cattle (UCAPS)'* [23], *'Posture Scoring System (PSS)'* [17], *'Multidimensional Pain Scoring System (MPSS)'* [49], *'Escala Composta Análogo-Visual (EA)'* [50], *'Veterinarian Pain Scale (VPS)'* [51] and *'Technician Pain Scale (TPS)'* [51]; three for ovine: *'Pain Scoring System for Ventricular Assist Devices-Implanted Sheep (PSS-VADS)'* [52], *'Behavior Assessment Scheme (BAS)'* [53], *'Unesp-Botucatu Composite Scale to Assess Acute Postoperative Abdominal Pain In Sheep (USAPS)'* [26]; and three for porcine: *'Unesp-Botucatu Pig Composite Pain Scale (UPAPS)'* [22], *'Perception of Pain, Distress and Discomfort Assessment (PDD)'* [54] and *'Behavioral Pain Scale in Piglets (BPSP)'* [55]. There were seven facial expression/grimace scales including one for bovine: *'Pain Assessment Based on Facial Expression (PABFE)'* [56]; three for ovine: *'Sheep Pain Facial Expression Scale (SPFES)'* [13], *'Sheep Grimace Scale (SGS)'* [24] and *'Lamb Grimace Scale (LGS)'* [34]; and three for porcine: *'Piglet Grimace Scale—A (PGS-A)'* [14], *'Piglet Grimace Scale—B (PGS-B)'* [57], and *'Sow Facial Expression Scale (SFES)'* [58]. The *'Cow Pain Scale (CPS)'* [35] is composed by facial expressions and behaviors for bovine (dairy cattle).

Part A 'Scale development' was not evaluated in four studies [59–62] because this information was not always provided (i.e. there was a second publication about a specific instrument on which the scale development had been reported in a first publication). Part A 'Measurement properties' was not evaluated in three instruments either because final scores could not be calculated (BAS [53] and SFES [57]) or when it was the case for a pilot study (PGS-A [14]).

The UCAPS [23], UPAPS [22] and USAPS [26] (n = 3), respectively for cattle, pigs and sheep, presented overall 'high' strength of evidence. The SPFES [13], LGS [34], PGS-B [57] and MPSS [49] (n = 4), respectively for sheep, lamb, piglets and dairy cattle, presented overall 'moderate' strength of evidence. The BPSP [55], VPS [51], TPS [51], CPS [35], SGS [24], SFES [58] and PABFE [56] (n = 7), respectively for piglets, cattle, cattle, cattle, sheep, sows and cattle, presented overall 'low' strength of evidence. The PSS [17], PGS-A [14], EA [50], PSS-VADS [52], BAS [53] and PDD [54] (n = 6), respectively for dairy cattle, piglets, cattle, sheep, sheep and pigs, presented overall 'very low' evidence. The PGS-B [57] had more than two studies available [60–62]. Pain scoring instruments presented variable length (number of items/AU), methods of scoring (i.e. real-time scoring, image or video assessment) and methods of calculating the final score (Table 4). Table 5 summarizes the consensus scores for each instrument for 'methodological quality' (Part A), 'quality of the findings' (Part B) and 'quality of evidence' (Part C). The category 'content validity' was not rated for facial/grimace scales [45]. Table 6 presents the findings of measurement properties of instruments included in this systematic review. Detailed population characteristics for these studies are included in the supplementary material (S2 Table).

None of the studies reported instrument feasibility, time needed for completion of pain assessment, or if training was required for the use of the instrument. Three studies presented the end-user of the instrument: the VPS [51] for veterinarians, the TPS [51] for veterinary nurses/technicians and the PSS-VADS [52] for veterinarians, researchers, and animal care staff. Most instruments provided clear item descriptions, and some included a manual. The UCAPS [23], SPFES [13], SGS [24], LGS [34], SFES [58], PGS-A [14] and PGS-B [57, 60–62] provided images whereas the CPS [35] provided images and drawings of facial expressions. Additionally, the USAPS [26] and UPAPS [22] provided videos for each item/score of the scale.

## Discussion

This systematic review presents evidence relating to the measurement properties of 20 scoring instruments used for pain assessment of bovine, ovine and porcine. Our results have identified

**Table 4. Summary of characteristics of pain scoring instruments in farm animals included in this systematic review.**

| Species / Scale [Ref] | Pain stimulus | Number of items or action units (AU) | Calculation method for final scores and cut-off score if available | Method of scoring (original) / alternative [ref] |
|---|---|---|---|---|
| **Bovine / UCAPS** [23] | Castration | 5 items—Locomotion, Activity, Appetite, Interactive Behavior, Miscellaneous Behaviors | 10 (sum) Cut-off: 4 out of 10 [23] or 3 out of 10 [59] | Video and RT scoring [23] / video [59] |
| **Bovine / PSS** [17] | Lameness | 6 items—Overall Locomotion Assessment, Spine Curvature, Speed, Tracking, Head Carriage, Abduction / Adduction | Final score not calculated (each item is scored from 1 to 5) | RT scoring |
| **Bovine / MPSS** [49] | Mastitis | 8 items—General Subjective Assessment, Postural Behavior, Interactive Behavior, Response to Food, Sacrum Position, Reaction to Back Palpation, Udder Edema, Udder Palpation | 42 (sum) | RT scoring |
| **Bovine / EA (Escala Composta Análogo-Visual)** [50] | Castration | 7 items—Respiratory Rate, Agitation, Appetite / Rumination, Posture, Contract Abdomen, Facial Expression of Pain, Auto-Auscultation | 15 (sum) | Video |
| **Bovine / VPS** [51] | Rumenotomy (left flank laparotomy) | 9 items—Temperature, Heart Rate, Respiratory Rate, and Mean Arterial Blood Pressure Recording, Interactive Behavior Attention, Response to Withers Pinch, Well-being, Appetite, Facial Expression, Posture | 25 (sum) | RT scoring |
| **Bovine / TPS** [51] | Rumenotomy (left flank laparotomy) | 8 items—Not Approaching Food, Not Eating, Not Ruminating, Abnormal Posture, Unusual Behavior when close to the Observer, Fear OR Avoidance, Vocalization OR Teeth Grinding, Aggressiveness | 8 (sum) | RT scoring |
| **Bovine / CPS** [35] | Clinical pain | 6 items—Attention Towards the Surroundings, Head Position, Ear Position, Facial Expression, Response to Approach, Back Position | 10 (sum) Cut-off: 3 out of 10 | RT scoring |
| **Bovine / PABFE** [56] | Castration | 6 AU—Reactivity, Vocalization, Muzzle, Mouth, Eye, Above the Eye | 6 (sum) | Image (screenshots from videos) |
| **Ovine / SPFES** [13] | Footrot and mastitis | 5 AU—Orbital Tightening, Cheek (Masseter) Tightening, Ear Position, Lip and Jaw Profile, Nostril, Philtrum Shape | 10 (sum) | Image (photographs) |
| **Ovine / PSS-VADS** [52] | Thoracotomy for surgical implantation of an infant ventricular assist device | 10 items—Posture, Restlessness, Heart Rate, Respiratory Rate, Pain on Palpation of Surgical Site, Kicking at Abdomen or Stomping Feet, Excessive Vocalization, Bruxism, Mental Status, Eating, Drinking | 25 (sum) Cut-off: 3–9 out of 25 | RT scoring |
| **Ovine / BAS*** [53] | Castration with the device Burdizzo | 9 items—General Attitude, Ear Position, Position of the Eyelid, Other Facial Expressions, Standing Postures, Lying Postures, Postures of the Legs, Clinical Signs, Abnormal Activities | Final score not calculated (each item is scored differently) | RT scoring |
| **Ovine / SGS** [24] | Unilateral osteotomy (right tibia) | 3 AU—Orbital Tightening, Ear and Head Position, Flehmen response | 7 (sum) | Image (screenshots from videos) |
| **Ovine / LGS** [34] | Tail-docking | 5 AU—Orbital Tightening, Nose Features, Mouth Features, Cheek Flattening, Ear Posture | 2 (average) | Image (screenshots from videos) |
| **Ovine / USAPS** [26] | Elective laparoscopy | 6 items—Interaction, Locomotion, Head Position, Posture, Activity, Appetite | 12 (sum) Cut-off: 4 out of 12 | Video |
| **Porcine / UPAPS** [22] | Castration | 6 items—Attention to Affected Area, Locomotion, Activity, Appetite, Interactive Behavior, Miscellaneous Behaviors | 18 (sum) Cut-off: 6 out of 18 | Video |
| **Porcine / PGS-A*** [14] | Castration and tail docking | 7 AU—Temporal Tension, Forehead Profile, Orbital Tightening, Cheek Tension, Tension Above Eyes, Snout Plate Changes, Snout Angle | Final score not calculated (each AU is scored independently) | Image (screenshots from videos) |
| **Porcine / PGS-B** [57] | Castration and tail docking | 3 AU—Ear Position, Cheek Tightening / Nose Bulge, Orbital Tightening | 5 (sum) | Image (screenshots from videos)/[60–62] |
| **Porcine / SFES*** [58] | Farrowing (sow parturition) | 5 AU—Tension Above Eyes, Snout Angle, Neck Tension, Temporal Tension, Ear Position | Not reported | Image (screenshots from videos) |

(*Continued*)

**Table 4.** (Continued)

| Species / Scale [Ref] | Pain stimulus | Number of items or action units (AU) | Calculation method for final scores and cut-off score if available | Method of scoring (original) / alternative [ref] |
|---|---|---|---|---|
| **Porcine / PDD** [54] | Lameness and rectal prolapse | 5 items—Unprovoked Behavior, Behavioral Responses to External Stimuli, Appearance, Body Condition Score, Clinical Signs | 20 (sum + 1 bonus per item) | RT scoring |
| **Porcine / BPSP** [55] | Castration | 22 items associated with how a piglet reacts/vocalizes during surgery or sprinkling of a topical product | 28.93 (sum) | Information not available |

Ref: Reference number between brackets. AU: Action Units. RT: Real-time method of scoring. UCAPS: Unesp-Botucatu Unidimensional Composite Pain Scale for assessing postoperative pain in cattle. PSS: Posture Scoring System. MPSS: Multidimensional Pain Scoring System. EA: Escala Composta Análogo-Visual. VPS: Veterinarian Pain Scale. TPS: Technician Pain Scale. CPS: Cow Pain Scale. PABFE: Pain Assessment Based on Facial Expression. SPFES: Sheep Pain Facial Expression Scale. PSS-VADS: Pain Scoring System for Ventricular Assist Devices-Implanted Sheep. BAS: Behavior Assessment Scheme. SGS: Sheep Grimace Scale. LGS: Lamb Grimace Scale. USAPS: Unesp-Botucatu Composite Scale to Assess Acute Postoperative Abdominal Pain in Sheep. UPAPS: Unesp-Botucatu Pig Composite Pain Scale. PGS-B: Piglet Grimace Scale-b. PGS-A: Piglet Grimace Scale-a. SFES: Sow Facial Expression Scale. PDD: Perception of Pain, Distress and Discomfort Assessment. BPSP: Behavioral Pain Scale in Piglets.

*For instruments scored only for Part A1 (development).

Note: Data retrieved from the articles included in this systematic review and reported herein are subject to bias or error attributable to any misinterpretation or unclear reporting of the results.

the strength and weakness of evidence related to pain scoring instruments revealing potential targets for future research with the ultimate benefit of improving animal welfare.

The majority of pain scoring instruments presented overall 'low' and 'very low' strength of evidence [14, 17, 24, 35, 50–56, 58] due to a small number of studies available, inadequate methodological quality, and/or conflicting or indeterminate quality of findings according to the COSMIN guidelines [38]. On the other hand, the UCAPS [23], UPAPS [22] and USAPS [26] presented with overall 'high' strength of evidence as studies showed robust and thorough statistical approach for scale development and validity of measurement properties. In this case, low ratings are potentially related to the rigorous of the COSMIN guidelines since the final score for each category is the lowest score from all criteria within that category. In other words, regardless of how many 'very good' or 'moderate' ratings a study received for different criteria, the rating would be 'low' if one of these criteria was scored as 'low'.

Content validity determines the degree to which the content of an instrument is an adequate reflection of the construct to be measured [46] (e.g. pain). It consists of a judgement whether the instrument presents relevant content or domains [36, 63]. The UCAPS [23], USAPS [26], and UPAPS [22] presented a 'high' strength of evidence for this measurement property with reported content validity index based on expert analysis, development of ethogram and literature findings [26, 64]. The COSMIN guidelines do not specify the number of experts required for content validity during scale development (Tables 1 and 2). However, it has been suggested that a minimum of four to five experts should be adequate for initial content validation [36] as used in the above scales with 'high' strength of evidence. The CPS [35] presented 'moderate' strength of evidence using expert opinion without calculating the content validity index. Most other instruments scored 'very low' [17,49–55] as they presented inadequate or unclear content validity.

Internal consistency describes the average correlations among items/AU of the instrument using the Cronbach's alpha, Kuder–Richardson or split halves [36]. The Cronbach's alpha coefficient interpretation is commonly used and classified as follows: > 0.80 (excellent), 0.75–0.80 (very good), 0.70–0.74 (good), 0.65–0.69 (acceptable) and 0.60–0.64 (minimally acceptable) [65]. For most instruments, internal consistency was not reported or performed [17, 24,

**Table 5. Summary of the consensus scores for each pain scoring instrument in farm animals regarding assessment of methodological quality (Part A), quality of the findings (Part B) and quality of evidence (Part C), according the order of analysis.**

| Species / Scale [ref] | Category | Total number of studies | Part A (methodological quality: number of studies) | Part B (overall quality of findings) | Part C (overall strength of evidence) | Final Overall Evidence |
|---|---|---|---|---|---|---|
| **Bovine / UCAPS** [23, 59] | General design requirements and relevance | 1 | A:1 | + | Moderate | High |
| | Content validity and comprehensibility | 1 | V:1 | + | High | |
| | Internal consistency | 2 | V:2 | + | High | |
| | Reliability | 2 | A:2 | +/- | Moderate | |
| | Measurement error | 2 | V:1 D:1 | + | High | |
| | Criterion validity | 2 | A:2 | + | Moderate | |
| | Construct validity | 2 | V:2 | + | High | |
| | Responsiveness | 2 | V:2 | + | High | |
| | Cross-cultural validity | 1 | V:1 | ? | High | |
| **Bovine / PSS** [17] | General design requirements and development | 1 | D:1 | ? | Very Low | Very Low |
| | Content validity and comprehensibility | 1 | I:1 | ? | Very Low | |
| | Internal consistency | 0 | N | ? | Unknown | |
| | Reliability | 1 | I:1 | ? | Very Low | |
| | Measurement error | 0 | N | ? | Unknown | |
| | Criterion validity | 1 | A:1 | ? | Low | |
| | Construct validity | 1 | I:1 | ? | Very Low | |
| | Responsiveness | 0 | N | ? | Unknown | |
| | Cross-cultural validity | 0 | N | ? | Unknown | |
| **Bovine / MPSS** [49] | General design requirements and development | 1 | D:1 | + | Low | Moderate |
| | Content validity and comprehensibility | 1 | I:1 | ? | Very Low | |
| | Internal consistency | 0 | N | ? | Unknown | |
| | Reliability | 0 | N | ? | Unknown | |
| | Measurement error | 0 | N | ? | Unknown | |
| | Criterion validity | 1 | A:1 | + | Moderate | |
| | Construct validity | 1 | V:1 | + | High | |
| | Responsiveness | 0 | N | ? | Unknown | |
| | Cross-cultural validity | 0 | N | ? | Unknown | |
| **Bovine / EA** [50] | General design requirements and development | 1 | D:1 | + | Low | Very Low |
| | Content validity and comprehensibility | 1 | I:1 | ? | Very Low | |
| | Internal consistency | 0 | N | ? | Unknown | |
| | Reliability | 1 | I:1 | ? | Very Low | |
| | Measurement error | 0 | N | ? | Unknown | |
| | Criterion validity | 1 | A:1 | - | Moderate | |
| | Construct validity | 1 | I:1 | + | Very Low | |
| | Responsiveness | 0 | N | ? | Unknown | |
| | Cross-cultural validity | 0 | N | ? | Unknown | |

(*Continued*)

**Table 5.** (Continued)

| Species / Scale [ref] | Category | Total number of studies | Part A (methodological quality: number of studies) | Part B (overall quality of findings) | Part C (overall strength of evidence) | Final Overall Evidence |
|---|---|---|---|---|---|---|
| **Bovine / VPS** [51] | General design requirements and development | 1 | D:1 | + | Low | Low |
| | Content validity and comprehensibility | 1 | I:1 | ? | Very Low | |
| | Internal consistency | 1 | V:1 | - | High | |
| | Reliability | 0 | N | ? | Unknown | |
| | Measurement error | 0 | N | ? | Unknown | |
| | Criterion validity | 0 | N | ? | Unknown | |
| | Construct validity | 1 | D:1 | ? | Very Low | |
| | Responsiveness | 0 | N | ? | Unknown | |
| | Cross-cultural validity | 0 | N | ? | Unknown | |
| **Bovine / TPS** [51] | General design requirements and development | 1 | D:1 | + | Low | Low |
| | Content validity and comprehensibility | 1 | I:1 | ? | Very Low | |
| | Internal consistency | 1 | V:1 | + | High | |
| | Reliability | 0 | N | ? | Unknown | |
| | Measurement error | 0 | N | ? | Unknown | |
| | Criterion validity | 0 | N | ? | Unknown | |
| | Construct validity | 1 | D:1 | ? | Very Low | |
| | Responsiveness | 0 | N | ? | Unknown | |
| | Cross-cultural validity | 0 | N | ? | Unknown | |
| **Bovine / CPS** [35] | General design requirements and development | 1 | D:1 | + | Low | Low |
| | Content validity and comprehensibility | 1 | A:1 | + | Moderate | |
| | Internal consistency | 0 | N | ? | Unknown | |
| | Reliability | 1 | D:1 | - | Low | |
| | Measurement error | 1 | A:1 | ? | Low | |
| | Criterion validity | 0 | N | ? | Unknown | |
| | Construct validity | 1 | D:1 | + | Low | |
| | Responsiveness | 0 | N | ? | Unknown | |
| | Cross-cultural validity | 0 | N | ? | Unknown | |
| **Bovine / PABFE** [56] | General design requirements and development | 1 | D:1 | + | Low | Low |
| | Content validity and comprehensibility | 0 | N | ? | Unknown | |
| | Internal consistency | 1 | A:1 | ? | Low | |
| | Reliability | 1 | D:1 | +/- | Very Low | |
| | Measurement error | 0 | N | ? | Unknown | |
| | Criterion validity | 0 | N | ? | Unknown | |
| | Construct validity | 0 | N | ? | Unknown | |
| | Responsiveness | 0 | N | ? | Unknown | |
| | Cross-cultural validity | 0 | N | ? | Unknown | |

(*Continued*)

**Table 5.** (Continued)

| Species / Scale [ref] | Category | Total number of studies | Part A (methodological quality: number of studies) | Part B (overall quality of findings) | Part C (overall strength of evidence) | Final Overall Evidence |
|---|---|---|---|---|---|---|
| Ovine / SPFES [13] | General design requirements and development | 1 | D:1 | + | Low | Moderate |
| | Content validity and comprehensibility | 0 | N | ? | Unknown | |
| | Internal consistency | 1 | A:1 | ? | Low | |
| | Reliability | 1 | A:1 | + | Moderate | |
| | Measurement error | 1 | A:1 | + | Moderate | |
| | Criterion validity | 1 | D:1 | - | Low | |
| | Construct validity | 1 | A:1 | + | Moderate | |
| | Responsiveness | 1 | A:1 | + | Moderate | |
| | Cross-cultural validity | 0 | N | ? | Unknown | |
| Ovine / PSS-VADS [52] | General design requirements and development | 1 | D:1 | - | Low | Very Low |
| | Content validity and comprehensibility | 1 | I:1 | ? | Very Low | |
| | Internal consistency | 0 | N | ? | Unknown | |
| | Reliability | 0 | N | ? | Unknown | |
| | Measurement error | 0 | N | ? | Unknown | |
| | Criterion validity | 0 | N | ? | Unknown | |
| | Construct validity | 1 | I:1 | ? | Very Low | |
| | Responsiveness | 0 | N | ? | Unknown | |
| | Cross-cultural validity | 0 | N | ? | Unknown | |
| Ovine / BAS* [53] | General design requirements and development | 1 | D:1 | + | Low | Very Low |
| | Content validity and comprehensibility | 1 | I:1 | ? | Very Low | |
| Ovine / SGS [24] | General design requirements and development | 1 | D:1 | ? | Very Low | Low |
| | Content validity and comprehensibility | 0 | N | ? | Unknown | |
| | Internal consistency | 0 | N | ? | Unknown | |
| | Reliability | 1 | A:1 | + | Moderate | |
| | Measurement error | 1 | D:1 | ? | Very Low | |
| | Criterion validity | 1 | I:1 | - | Very Low | |
| | Construct validity | 1 | V:1 | + | High | |
| | Responsiveness | 0 | N | ? | Unknown | |
| | Cross-cultural validity | 0 | N | ? | Unknown | |
| Ovine / LGS [34] | General design requirements and development | 1 | D:1 | + | Low | Moderate |
| | Content validity and comprehensibility | 0 | N | ? | Unknown | |
| | Internal consistency | 0 | N | ? | Unknown | |
| | Reliability | 1 | V:1 | - | High | |
| | Measurement error | 0 | N | ? | Unknown | |
| | Criterion validity | 0 | N | ? | Unknown | |
| | Construct validity | 1 | V:1 | + | High | |
| | Responsiveness | 0 | N | ? | Unknown | |
| | Cross-cultural validity | 0 | N | ? | Unknown | |

(*Continued*)

**Table 5.** (Continued)

| Species / Scale [ref] | Category | Total number of studies | Part A (methodological quality: number of studies) | Part B (overall quality of findings) | Part C (overall strength of evidence) | Final Overall Evidence |
|---|---|---|---|---|---|---|
| **Ovine / USAPS** [26] | General design requirements and development | 1 | A:1 | + | Moderate | High |
| | Content validity and comprehensibility | 1 | V:1 | + | High | |
| | Internal consistency | 1 | V:1 | + | High | |
| | Reliability | 1 | A:1 | +/- | Low | |
| | Measurement error | 1 | V:1 | + | High | |
| | Criterion validity | 1 | A:1 | + | Moderate | |
| | Construct validity | 1 | V:1 | + | High | |
| | Responsiveness | 1 | V:1 | + | High | |
| | Cross-cultural validity | 0 | N | ? | Unknown | |
| **Porcine / UPAPS** [22] | General design requirements and development | 1 | A:1 | + | Moderate | High |
| | Content validity and comprehensibility | 1 | V:1 | + | High | |
| | Internal consistency | 1 | V:1 | + | High | |
| | Reliability | 1 | A:1 | +/- | Low | |
| | Measurement error | 1 | V:1 | + | High | |
| | Criterion validity | 1 | A:1 | + | Moderate | |
| | Construct validity | 1 | V:1 | + | High | |
| | Responsiveness | 1 | V:1 | + | High | |
| | Cross-cultural validity | 0 | N | ? | Unknown | |
| **Porcine / PGS-B** [57, 60–62] | General design requirements and development | 1 | D:1 | + | Low | Moderate |
| | Content validity and comprehensibility | 0 | N | ? | Unknown | |
| | Internal consistency | 0 | N | ? | Unknown | |
| | Reliability | 2 | D:1 I:1 | - | Very Low | |
| | Measurement error | 0 | N | ? | Unknown | |
| | Criterion validity | 1 | D:1 | - | Low | |
| | Construct validity | 4 | V:1 A:2 D:1 | + | High | |
| | Responsiveness | 3 | A:2 D:1 | ? | Moderate | |
| | Cross-cultural validity | 0 | N | ? | Unknown | |
| **Porcine / PGS-A*** [14] | General design requirements and development | 1 | I:1 | - | Very Low | Very Low |
| | Content validity and comprehensibility | 0 | N | ? | Unknown | |
| **Porcine / SFES*** [58] | General design requirements and development | 1 | D:1 | + | Low | Low |
| | Content validity and comprehensibility | 0 | N | ? | Unknown | |

(*Continued*)

**Table 5.** (Continued)

| Species / Scale [ref] | Category | Total number of studies | Part A (methodological quality: number of studies) | Part B (overall quality of findings) | Part C (overall strength of evidence) | Final Overall Evidence |
|---|---|---|---|---|---|---|
| **Porcine / PDD** [54] | General design requirements and development | 1 | D:1 | + | Low | Very Low |
| | Content validity and comprehensibility | 1 | I:1 | ? | Very Low | |
| | Internal consistency | 0 | N | ? | Unknown | |
| | Reliability | 1 | I:1 | + | Very Low | |
| | Measurement error | 0 | N | ? | Unknown | |
| | Criterion validity | 1 | A:1 | +/- | Low | |
| | Construct validity | 1 | I:1 | + | Very Low | |
| | Responsiveness | 0 | N | ? | Unknown | |
| | Cross-cultural validity | 0 | N | ? | Unknown | |
| **Porcine / BPSP** [55] | General design requirements and development | 1 | D:1 | ? | Very Low | Low |
| | Content validity and comprehensibility | 1 | I:1 | ? | Very Low | |
| | Internal consistency | | V:1 | + | High | |
| | Reliability | 0 | N | ? | Unknown | |
| | Measurement error | 0 | N | ? | Unknown | |
| | Criterion validity | 1 | D:1 | - | Low | |
| | Construct validity | 1 | I:1 | ? | Very Low | |
| | Responsiveness | 0 | N | ? | Unknown | |
| | Cross-cultural validity | 0 | N | ? | Unknown | |

Ref: Reference number between brackets. UCAPS: Unesp-Botucatu Unidimensional Composite Pain Scale for assessing postoperative pain in cattle. MPSS: Multidimensional Pain Scoring System. EA: Escala Composta Análogo-Visual. VPS: Veterinarian Pain Scale. TPS: Technician Pain Scale. CPS: Cow Pain Scale. PABFE: Pain Assessment Based on Facial Expression. SPFES: Sheep Pain Facial Expression Scale. PSS-VADS: Pain Scoring System for Ventricular Assist Devices-Implanted Sheep. BAS: Behavior Assessment Scheme. SGS: Sheep Grimace Scale. LGS: Lamb Grimace Scale. USAPS: Unesp-Botucatu Composite Scale to Assess Acute Postoperative Abdominal Pain in Sheep. UPAPS: Unesp-Botucatu Pig Composite Pain Scale. PGS-B: Piglet Grimace Scale-b. PGS-A: Piglet Grimace Scale-a. SFES: Sow Facial Expression Scale. PDD: Perception of Pain, Distress and Discomfort Assessment. BPSP: Behavioral Pain Scale in Piglets. Part A—Methodological quality: 'V' (very good), 'A' (adequate), 'D' (doubtful), 'I' (inadequate) or 'N' (not applicable / not reported). Part B—Quality of findings: '+' (sufficient or positive), '-' (insufficient or negative), '+/-' (inconsistent / conflicting findings), or '?' (indeterminate). Part C—Overall strength of evidence: High, Moderate, Low, Very Low or Unknown. Note: Data retrieved from the articles included in this systematic review and reported herein are subject to bias or error attributable to any misinterpretation or unclear reporting of the results.

35, 49, 50, 52, 54, 57, 58, 60–62]. This measurement property scored 'high' strength of evidence for the UCAPS [23, 59], USAPS [26], UPAPS [22], BPSP [55], TPS and VPS [51] using the Cronbach's alpha coefficient. Internal consistency was reported for two facial-based scales: the PABFE [56], in which the correlation between each AU and the sum of the AUs were evaluated and the SPFES [13], in which the same approach was used without coefficient reporting. Internal consistency indicates the interrelatedness of scale items or AUs. For example, the Cronbach alpha coefficient can be calculated by excluding each scale item. Increased alpha values indicate that the scale homogeneity is increased when excluding an item. The item-total correlation is also used for internal consistency to determine if an item is consistent with the others of the scale or the averaged measure [36].

Intra and inter-reliability are usually carried out using the intra-class correlation coefficient (ICC) or Kappa coefficient [36, 38, 66]. The Kendall's index of concordance uses ranks to assess the agreement between observers and was reported for the LGS [67]. Inter-rater

**Table 6. Summary of the findings of measurement properties for each pain scoring instrument in farm animals included in the systematic review.**

| Species / Scale [Ref] | Validity | | Reliability | | Responsiveness (treatments) | Internal consistency | Measurement error (sensitivity, specificity, or accuracy) | Observations |
|---|---|---|---|---|---|---|---|---|
| | Construct measured | Criterion (comparator; coefficient) | Inter-rater (coefficient / raters) | Intra-rater (coefficient / interval) | | | | |
| **Bovine / UCAPS** [23, 59] | Castration | VAS r = 0.839 NRS; r = 0.883 SDS; r = 0.866 [23] VAS; rho = 0.842 NRS; rho = 0.889 SDS; rho = 0.880 [59] | ICC = 0.52–0.80 for each individual item / 4 raters [a] [23] ICC only for items = 0.37–0.79 / 5 raters [b] [59] | ICC = 0.61–0.96 for each individual item / 1 month interval [23] ICC only for items = 0.65–0.87 rater 1 0.56–0.91 rater 2 0.68–0.88 rater 3 0.34–0.83 rater 4 / 1 month interval [59] | NSAID, OP | Cronbach's α = 0.86 [23] Cronbach's α = 0.82 [59] | Accuracy = 0.963 (AUC) [23] Accuracy = 0.983 (AUC) [59] | [a] 4 raters (3 blinded and 1 in-person evaluation) [b] 5 blinded raters (4 Italians and the original researcher) |
| **Bovine / PSS** [17] | Lameness | NR | Number of scores in agreement for items only = 17–40% / number of raters not reported | Number of scores in agreement only for items = 43–72% / same day assessed | NR | NR | NR | |
| **Bovine / MPSS** [49] | Mastitis | VAS; rho = 0.817 | NR | NR | NSAID | NR | NR | |
| **Bovine / EA** [50] | Castration | Cortisol; rho = 0.15 | Agreement > 90% / 3 raters [c] | NR | NSAID | NR | NR | [c] 3 blinded raters |
| **Bovine / VPS** [51] | Rumenotomy | NR | NR | NR | NSAID, OP | Cronbach's α = 0.67 | NR | |
| **Bovine / TPS** [51] | Rumenotomy | NR | NR | NR | NSAID, OP | Cronbach's α = 0.71 | NR | |
| **Bovine / CPS** [35] | Clinical pain | NR | Weighted kappa = 0.62 / 2 raters [d] | NR | NSAID | NR | Balanced accuracy = 0.71 | [d] 1 experienced rater, 1 inexperienced |
| **Bovine / PABFE** [56] | Castration | NR | NR | weighted kappa = 0.64–1.00 / time interval not reported [e] | NSAID | NR | NR | [e] 1 experienced rater |
| **Ovine / SPFES** [13] | Footrot and mastitis | Lameness; rho = 0.56 Lesion score; rho = 0.54 | ICC = 0.86 / 5 raters | NR | NSAID, ATB | AU correlates with others and total score (no coefficient reported) | Accuracy = 84% (global evaluation) | |
| **Ovine / SGS** [24] | Unilateral osteotomy | Clinical severity score; r = 0.47 [f] | ICC = 0.92 / 6 raters | NR | NSAID, OP | NR | Accuracy = 68.2% | [f] Correlation not performed at same time |
| **Ovine / LGS** [34] | Tail-docking | NR | W = 0.6–0.66[g] / 5 raters | NR | NR | NR | NR | [g] W = Kendall's index of concordance |
| **Ovine / USAPS** [26] | Elective laparoscopy | NRS; rho = 0.83 SDS; rho = 0.81 VAS; rho = 0.81 Facial scale; rho = 0.48 | ICC > 0.50 (0.53–0.74) / 4 raters [h] | ICC = 0.77 rater 1 0.84 rater 2 0.65 rater 3 0.72 rater 4 / 1 month interval | NSAID, OP | Cronbach's α = 0.81 | Accuracy = 0.953 (AUC) | [h] 4 blinded raters |

*(Continued)*

**Table 6.** (Continued)

| Species / Scale [Ref] | Validity | | Reliability | | Responsiveness (treatments) | Internal consistency | Measurement error (sensitivity, specificity, or accuracy) | Observations |
|---|---|---|---|---|---|---|---|---|
| | Construct measured | Criterion (comparator; coefficient) | Inter-rater (coefficient / raters) | Intra-rater (coefficient / interval) | | | | |
| **Porcine / UPAPS** [22] | Castration | VAS; rho = 0.846 NRS; rho = 0.878 SDS; rho = 0.854 | Weighted kappa [i] = 0.81 rater 1 0.80 rater 2 0.62 rater 3 / 3 raters | ICC = 0.88 (gold standard rater) 0.85 rater 1 0.79 rater 2 0.82 rater 3 / 1 month interval | NSAID, OP | Cronbach's α = 0.89 | Accuracy = 0.98 (AUC) | [i] Gold standard rater versus three others—4 blinded raters, 2 females and 2 males |
| **Porcine / PGS-B** [57, 60–62] | Castration and tail docking | General behaviors (active and inactive); r = -0.22 to 0.22 | ICC = 0.57 / 2 raters [57] ICC = 0.87 / 3 raters [60] | NR | NSAID, LA, OP | NR | NR | |
| **Porcine / PDD** [54] | Lameness and rectal prolapse | Lameness; rho = 0.980 Prolapse length; rho = 0.903 CRP; rho = 0.740 Cortisol; rho = 0.577 | ICC = 0.893 / 3 raters [j] | LOA = -4.56 to 4.96 / 3 hours interval | NR | NR | NR | [j] 2 females and 1 male rater |
| **Porcine / BPSP** [55] | Castration | Cortisol; Linear correlation coefficient = 0.36 (0.15–0.54) | NR | NR | NR | Cronbach's α = 0.88 | NR | |

Ref: Reference number between brackets. UCAPS: Unesp-Botucatu Unidimensional Composite Pain Scale for assessing postoperative pain in cattle. MPSS: Multidimensional Pain Scoring System. EA: Escala Composta Análogo-Visual. VPS: Veterinarian Pain Scale. TPS: Technician Pain Scale. CPS: Cow Pain Scale. PABFE: Pain Assessment Based on Facial Expression. SPFES: Sheep Pain Facial Expression Scale. PSS-VADS: Pain Scoring System for Ventricular Assist Devices-Implanted Sheep. BAS: Behavior Assessment Scheme. SGS: Sheep Grimace Scale. LGS: Lamb Grimace Scale. USAPS: Unesp-Botucatu Composite Scale to Assess Acute Postoperative Abdominal Pain in Sheep. UPAPS: Unesp-Botucatu Pig Composite Pain Scale. PGS-B: Piglet Grimace Scale-b. PGS-A: Piglet Grimace Scale-a. SFES: Sow Facial Expression Scale. PDD: Perception of Pain, Distress and Discomfort Assessment. BPSP: Behavioral Pain Scale in Piglets. AU: Action units. AUC: Area under the curve LOA: Limits of agreement. ICC: Intra-class correlation coefficient. r: Pearson's correlation coefficient. rho: Spearman's correlation coefficient. NR: Not reported. Treatments—OP: Opioids, NSAID: Non-steroidal anti-inflammatory drugs, LA: local anesthetics, ATB: Antibiotics. CRP: C-Reactive Protein. Note: Data retrieved from the articles included in this systematic review and reported herein are subject to bias or error attributable to any misinterpretation or unclear reporting of the results. Superscript letters (a-j) link observations to specific measurement properties of pain scoring instruments within the same line.

reliability was reported for ten instruments mostly by ICC [13, 22–24, 26, 54, 59] or weighted Kappa [22, 35, 57, 60]. Intra-rater reliability was reported for six instruments [17, 22, 23, 26, 54, 56, 59]. The interval between assessments ranged from three hours to 30 days. Intervals shorter than one week were considered inadequate as results could have been biased by memorization [36]. The LGS [57] was the only one instrument with a 'high' strength of evidence for reliability. Most of the instruments scored 'low' or 'very low' [17, 22, 63, 26, 35, 50, 54, 56, 57, 60, 61] due to inadequate design or unclear reporting of reliability testing. For example, for the UCAPS, UPAPS and USAPS did not receive high scores for reliability because methods for ICC calculation were not properly described. Future studies should focus on reliability reporting to improve the measurement properties of pain scoring instruments in farm animals. Additionally, results of reliability testing and other measurement properties could have been influenced by the sample size (i.e. number of animals included) among studies. Indeed, the COSMIN criteria do not take study sample size in consideration during methodological quality assessment.

Measurement error refers to accuracy, sensitivity and specificity of an instrument. Accuracy may vary according to the user of the scale. Only six instruments reported measurement error. The UCAPS [23, 59], USAPS [26] and UPAPS [22] presented overall 'high' strength of evidence. These studies used the Receiver Operating Characteristics (ROC) to determine sensitivity, specificity and accuracy, and calculate the area under the ROC curve [68]. The SPFES [13] also reported a ROC curve. However, it scored overall 'moderate' because only the scores of an experienced rater were considered and it was unclear if the rater also had participated in scale development. The CPS [35] and SGS [24] scored 'low' and 'very low', respectively, because a global judgment (absence or presence of pain) based on the rater's opinion was used to determine accuracy, which may be biased [47] and does not take into consideration the scores of the instruments.

The UCAPS [23, 59], USAPS [26] and UPAPS [22] reported a cut-off for analgesic administration using the ROC curve. Furthermore, the CPS [35] suggested a cut-off value for rescue analgesia based on the differences between clinical pain and control groups, whereas the PSS-VADS [52] empirically suggested a cut-off which was considered inadequate. Future studies should properly calculate the cut-off for analgesic intervention as it may guide clinical decision-making of veterinarians in practice improving welfare and ensuring that painful animals are properly treated.

Criterion validity reflects the degree to which the scores are an adequate reflection of a 'gold standard' or another previously validated method for measuring the same construct [46]. None of the scales presented 'high' strength of evidence for criterion validity as the presence of a 'gold standard' instrument is usually not available in veterinary medicine and unidimensional scales are used instead (i.e. VAS, NRS, SDS). The UCAPS [23, 59], MPSS [49], EA [50], USAPS [26] and UPAPS [22] presented 'moderate' strength of evidence with acceptable values for Spearman or Pearson's correlation as comparisons were performed with unidimensional pain scales which are not species-specific [22, 23, 26, 49, 59] or with cortisol concentrations that may be increased in acute pain [69, 70], but also due to stress. Five instruments scored 'low' as comparisons were performed with pain assessment methods considered to be inadequate: the PSS [17], SPFES [13], PGS-B [57, 60–62], PDD [54] and BPSP [55]. The SGS [24] scored 'very low' for criterion validity as the Pearson's correlation was < 0.5.

Construct validity measures the degree to which the scores of an instrument identify what is meant to [46] (i.e. discrimination between pain and pain-free states). This measurement property was reported in all studies except for the PABFE [56]. The UCAPS [23, 59], MPSS [49], SGS [24], LGS [57], USAPS [26], UPAPS [22], and PGS-B [57, 60–62] (n = 7) presented 'high' strength of evidence using surgical or clinical models of pain while in which pain scores were different between painful and pain-free animals or before and after surgery. The SPFES [13] scored 'moderate' as reporting for subgroups was unclear. The CPS [35] scored 'low' as it was unclear if the construct being evaluated was pain or disease. The other instruments (n = 7) scored 'very low' [17, 50–52, 54, 55] because construct validity was not reported, the statistical method was not appropriate, study design flaws were identified, or reporting of findings was unclear.

Responsiveness was considered when decreases in pain scores were statistically significant after analgesic intervention [45, 71]. The UCAPS [23, 59], USAPS [26], and UPAPS [22] presented 'high' strength of evidence with differences in pain scores after the administration of analgesics using non-steroidal anti-inflammatory drugs and opioids. The SPFES [13] and PGS-B [57, 60–62] presented 'moderate' strength of evidence. These instruments indeed had significant changes in pain scores in response to different analgesic interventions (e.g. non-steroidal anti-inflammatory drugs, local anesthetics, opioids). However, the description of the intervention was unclear (i.e. dose, route of administration, etc.) or the time interval between

administration of the intervention and pain scoring was not ideal [45]. Responsiveness was not assessed for the CPS [35]. Although response to analgesic administration was reported in the original study, this step was only performed during the development of the scale (n = 15 items) and not in the actual scale (n = 6 items) [35]. Most of the instruments did not report responsiveness [17, 24, 34, 49–52, 54–56] and this is also a critical measurement property to be addressed in future studies.

Cross-cultural validity assesses whether items of a translated or culturally adapted instrument properly reveal the originally developed instrument [46]. The only instrument subjected to cross-cultural validity was UCAPS [23]. It was first developed in Portuguese and had cross-cultural validation in Italian [59]. There is a need for further cross-cultural validity for farm animal pain assessment instruments when used in other languages due to semantic variations and the risk of 'lost in translation' when the original meaning is not reflected in the translated version.

This systematic review has limitations. The small number of studies for most instruments or unclear reporting may have reduced the overall strength of evidence of measurement properties of pain scoring instruments. The COSMIN checklists may be used as guidelines to circumvent some limitations in future studies planning to develop and validate pain scoring instruments to avoid inappropriate methodology. For example, none of the studies reported the interpretability and feasibility of these instruments and this is a major gap to be addressed in the future and during scale development. However, as mentioned before, low ratings for 'methodological quality' were potentially related to the rigor of the COSMIN guidelines since the final score for each category is the lowest score from all criteria within that category. Additionally, some items of the COSMIN methodology were adapted in this study to circumvent the limitations related to pain scoring instruments in individuals that cannot self-report pain and evaluations are performed by a proxy. Our methodology was strengthened by using a modified GRADE approach for grading the quality of evidence in systematic reviews of patient-reported outcome measures. However, the COSMIN guidelines appreciate that the methods for using GRADE require further validation; on the other hand, to the authors' knowledge, this is the only suitable method available for this type of grading. Finally, this systematic review did not assess the effects of the observer gender in the development and validation of pain scoring instruments. This issue was poorly reported in the studies included (Table 6; Observations) and this information is not required by the COSMIN. As previously described, female observers may have more empathy than male individuals during pain assessment [72]. It is not clear how this could affect scale development and validation—using different observers or those of the same gender, for example.

## Conclusions

This systematic review presents the evidence related to the measurement properties of pain scoring instruments in farm animals. A total of 20 pain scoring instruments for bovine, ovine, and porcine were selected, according to the inclusion criteria. The UCAPS, UPAPS and USAPS showed the highest overall strength of evidence. Instruments with overall 'moderate' strength of evidence included the MPSS, for bovine, the SPFES and LGS for ovine, and the PGS-B for porcine. Results for studies concerning the PSS, the EA, the VPS, the TPS, the CPS, the PABFE, the PSS-VADS, the BAS, the SGS, the PGS-A, the SFES, the PDD and the BPSP showed that future research is warranted to address the limitations of these pain scoring instruments. In the meantime, these pain scoring instruments should be used with caution with the understanding of their strengths and limitations as reported in this article. The most reported measurement property was construct validity, followed by criterion validity and

reliability. Internal consistency, measurement error and responsiveness have been understudied whereas 'cross-cultural validity' was performed for only one scale. This review identifies the gaps of knowledge with these instruments (low or very low strength of evidence due to small number of studies, inadequate methodology or design, conflicting or undetermined quality of findings or reporting; lack of cut-off for analgesic intervention; inappropriate comparisons for criterion validity, etc.), species that are lacking validated pain scoring instruments and potential targets for future studies in farm animals. Indeed, instruments with reported validation are urgently required for pain assessment of buffalos, goats, camels and avian species to provide tools to improve the welfare of these animals.

## Supporting information

**S1 Table. Detailed criteria used for assessing methodological quality of each included study.**
(DOCX)

**S2 Table. Summary of the population characteristics in the studies included in the systematic review.**
(DOCX)

**S1 Checklist.**
(DOCX)

## Acknowledgments

Ms. Marie-Claude Poirier for the invaluable help with databases, search terms and literature search.

## Author Contributions

**Conceptualization:** Paulo Vinícius Steagall.

**Investigation:** Rubia Mitalli Tomacheuski, Beatriz Paglerani Monteiro, Marina Cayetano Evangelista, Paulo Vinícius Steagall.

**Methodology:** Rubia Mitalli Tomacheuski, Beatriz Paglerani Monteiro, Marina Cayetano Evangelista, Stelio Pacca Loureiro Luna, Paulo Vinícius Steagall.

**Project administration:** Beatriz Paglerani Monteiro, Paulo Vinícius Steagall.

**Resources:** Marina Cayetano Evangelista, Stelio Pacca Loureiro Luna, Paulo Vinícius Steagall.

**Supervision:** Beatriz Paglerani Monteiro, Paulo Vinícius Steagall.

**Validation:** Beatriz Paglerani Monteiro, Marina Cayetano Evangelista.

**Visualization:** Rubia Mitalli Tomacheuski.

**Writing – original draft:** Rubia Mitalli Tomacheuski.

**Writing – review & editing:** Beatriz Paglerani Monteiro, Marina Cayetano Evangelista, Stelio Pacca Loureiro Luna, Paulo Vinícius Steagall.

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
