## [Decision Letter · Decision Letter 0]

24 Oct 2022

PONE-D-22-02200Measurement properties of pain scoring instruments in farm animals: a systematic review using the COSMIN checklist\\PLOS ONE

Dear Dr. Steagall,

Thank you for submitting your manuscript to PLOS ONE. After careful consideration, we feel that it has merit but does not fully meet PLOS ONE’s publication criteria as it currently stands. Therefore, we invite you to submit a revised version of the manuscript that addresses the points raised during the review process.

This systematic review has provided informative description of current research status of pain scales for animals. I found the reviewers' comments are very helpful for improving this manuscript. Please address them accordingly.

We also noticed you have some minor occurrence of overlapping text with the following previous publication(s), which needs to be addressed:

- Tomacheuski RM, Monteiro BP, Evangelista MC, Luna SPL, Steagall PV. Measurement properties of pain scoring instruments in farm animals: A systematic review protocol using the COSMIN checklist. PLoS One. 2021;16: e0251435. doi:10.1371/journal.pone.0251435

The text that needs to be addressed involves the Introduction section of your manuscript. In your revision ensure you cite all your sources (including your own works), and quote or rephrase any duplicated text outside the methods section. Further consideration is dependent on these concerns being addressed. 

We look forward to receiving your revised manuscript.

Kind regards,

Xiaodan Tang

Academic Editor

PLOS ONE

Journal Requirements:

Reviewers' comments:

Reviewer's Responses to Questions

**Comments to the Author**

1. Does the manuscript adhere to the experimental procedures and analyses described in the Registered Report Protocol?

If the manuscript reports any deviations from the planned experimental procedures and analyses, those must be reasonable and adequately justified.

Reviewer #1: Yes

Reviewer #2: Yes

2. If the manuscript reports exploratory analyses or experimental procedures not outlined in the original Registered Report Protocol, are these reasonable, justified and methodologically sound?

A Registered Report may include valid exploratory analyses not previously outlined in the Registered Report Protocol, as long as they are described as such.

Reviewer #1: Yes

Reviewer #2: No

3. Are the conclusions supported by the data and do they address the research question presented in the Registered Report Protocol?

The manuscript must describe a technically sound piece of scientific research with data that supports the conclusions. The conclusions must be drawn appropriately based on the research question(s) outlined in the Registered Report Protocol and on the data presented.

Reviewer #1: Yes

Reviewer #2: Yes

4. Have the authors made all data underlying the findings in their manuscript fully available?

Reviewer #1: No

Reviewer #2: Yes

5. Is the manuscript presented in an intelligible fashion and written in standard English?

Reviewer #1: Yes

Reviewer #2: Yes

6. Review Comments to the Author

Please use the space provided to explain your answers to the questions above. (Please upload your review as an attachment if it exceeds 20,000 characters)

Reviewer #1: Measurement properties of pain scoring instruments in farm animals: a systematic review using the COSMIN checklist.

This systematic review has investigated the measurement properties of pain scoring instruments intended for farm animals. The study reports in detail on the measurement properties of 20 tools for three species, cow, sheep and pig. The study also describes the nature and purpose of the properties. A concensus-based guideline made for selection of human health measurement instruments is used as a protocol.

This type of critical reviews of relatively new clinical methods are highly necessary and welcomed. The study is thorough and well planned. Despite that there are no original findings of the study, I find it very valuable for the future research. Not surprisingly, the study shows that some scales have better performance parameters than others. This probably mainly reflects the need of studies like this: a systematical approach to validation has not yet become a part of the pain scale developers mind sets.

My main critical comments concerns the rather un-reflected use of COSMIN – is it really that easy to apply a human tool intended for verbal self-report, to a situation where humans observe animals for pain? I would like a commentary on that in the Introduction or Discussion, where it fits best. In relation to that, since there is no gold standard for pain in animals, do the authors have any suggestions for modification of the guideline. No scale is better than its construct validity, and the discussion becomes rather technical, where it would be nice to get an impression of the consequences of certain poor or missing properties, if possible.

I below give a number of minor comments, some are merely typos.

Line 44: Most... Give number for clarity .

Line 47. Llama, Alpaca?

Line 57. ..Or low empathic capacity of farmers (barn blindness).

Line 66: Also, they do not necessarily measure the suffering component of pain.

Line 66. Other surrogate measures... lameness and activity are also surrogates?

Line 68... Are also not necessarily..

Line 80: what is meant by appearance here?

Line 83: What is meant by curved lips in a cow?

Line 92: Why is species specificity of importance?

Line 138: peer-reviewed?

Line 187: this would be helpful as supplementary material.

Line 197: Helpful as supplementary material.

Line 199-204: This section belongs to Introduction, where nothing is mentioned about COSMIN. As it is in the title, a small introduction would be helpful.

Table 2, 2nd last section. What do Action Units mean in this context?

Table 4: Last row. How can the study be included if scoring method is not available?

Line 299: there -instead of they?

Line 381. Content validity needs a more in depth discussion. As already mentioned, there is no gold standard for pain, and has consistently been shown that expert opinions differ much. Therefore, a more critical approach to content validity is warranted. Scales are no better than their content validity, despite excellent rater agreements and other criteria.

Line 389. You could add here that content validity of pain scales is a problem, since there is no gold standards to rely on. In addition, studies has shown considerable overlap between behaviours in pain and during stress, putting pressure on the correct identification of pain.

Line 399: Please explain the hypothesis behind why this is relevant. Is it the frequency of facial action units, which is meant? And why should that be correlated to the sum om AUs?

Line 401: Is it your opinion ICC is used correctly in all cited papers?

Line 407: So intra-rater agreement was always done on footage. Please explain how it should be done.

Line 423: The global judgement is used in many pain publications in order to avoid circularity because of including the items of the scoring.

Line 437. Using unidimensional scales does not make pain scoring better! The numbers looks good, but please explain why you think they can be used for criterion validity. If they were pain scales, we did not need to develop new scales.

Line 441: Weak argument, since cortisol concentrations are not specific to pain, as you already have mentioned.

Line 453: Yes, this is actually a good discussion. As you already mention, there is no way to measure the degree of pain experience, are high pain scores sign of high pain intensity, or high pain probability or both? Disease: we have a biological understanding of pain pathology, and pathology (inflammation, trauma, etc.) may be a good proxy for pain probability, just as we accept for example lameness as a pain proxy.

Line 457: Are pain scales linear? Which statistics should be used to show significance or what is meant by significant?

Line 461. I believe that study 35 was based on analgesic testing, this study is not mentioned.

Line 482. The rigidity of COSMIN. Yes. Could you discuss if any items should be omitted or modified for use in animal pain scale development?

Line 486: Could you discuss the consequences of using scales with different deficiencies?

Reviewer #2: Based on the COSMIN guidelines for assessing health measure instruments the present review seeks to provide evidence of reliability, validity and sensitivity of pain scoring instruments for farmed animals. The review identified 20 pain assessment protocols based on the initial search and following screening (inclusion and exclusion criteria). The steps of the COSMIN guidelines are followed nicely and result in a comprehensive overview of the 20 publications. The review adds important information to the sparse knowledge on validity within the growing field of pain scoring scales, as it identifies the most frequent shortcomings of publications on these instruments, provided helpful recommendations for future publications and further validation of pain scoring instruments.

The manuscript is well-written and easy to follow. However, there are some issues, I would like to address.

Firstly, I was a bit puzzled, that the complete introduction and the following parts all the way through to line 214 are exact copies of the previous report Tomacheuski et al. (2021)?

The materials and methods state, that no language restrictions were imposed. Based on this decision I wonder how correct translations were to be ensured and how evidence from journals were considered in this review? Were manuscripts eligible if they were not peer-reviewed?

The definition of farm animals in lines 186-195 might be nice to have already before search terms are listed.

Regarding the exclusion criteria, why was the sample size of the included studies not considered? This could also affect validity of study outcomes.

The results are nicely illustrated in the two tables, however, the figure 1 has a really low resolution in the pdf-version distributed.

Albeit the COSMIN approach ensures a good evaluation of the validity issues in regards to pain assessment protocols, the consequences of this rigid protocol are not really discussed in depth. Since most of the pain scoring instruments only are described by one publication (maximum four publications for the Porcine/PBS-B), there is actually not a lot of evidence yet. Hence, the discussion could be improved by adding a discussion of the included instruments' contents i.e. their feasibility and applicability and highlight their shortcomings in order to point out the remaining gaps of knowledge.

Additionally, a discussion of the definition of pain would be beneficial, as it is a subjective sensory and emotional experience, one could argue that it also depends on the level of empathy of the observer.

In case of expert opinion being used as to ensure content validity, the included studies ranking high on this measure did only consult 4 experts. How much evidence can four experts generate? How many experts would be required? I think this also needs to be addressed in the discussion.

Finally, in the conclusion it would be nice, if the 'gaps of knowledge' were described again to emphasize what needs to be considered when planning the development and publishing of new pain assessment methods.

7. PLOS authors have the option to publish the peer review history of their article (what does this mean?). If published, this will include your full peer review and any attached files.

Reviewer #1: No

Reviewer #2: No

---

## [Author Response · Author response to Decision Letter 0]

16 Nov 2022

Response to reviewers

PONE-D-22-02200

Measurement properties of pain scoring instruments in farm animals: a systematic review using the COSMIN checklist

PLOS ONE

Academic editor’s comments:

We also noticed you have some minor occurrence of overlapping text with the following previous publication(s), which needs to be addressed:

- Tomacheuski RM, Monteiro BP, Evangelista MC, Luna SPL, Steagall PV. Measurement properties of pain scoring instruments in farm animals: A systematic review protocol using the COSMIN checklist. PLoS One. 2021;16: e0251435. doi:10.1371/journal.pone.0251435

The text that needs to be addressed involves the Introduction section of your manuscript. In your revision ensure you cite all your sources (including your own works), and quote or rephrase any duplicated text outside the methods section. Further consideration is dependent on these concerns being addressed. 

Response: This has now been addressed with the editorial office and the academic editor. According to the journal’s guidelines, Registered Report Research Articles report the results of all planned analyses previously published in the journal (Tomacheuski et al. 2021) and, if relevant, detail and justify all deviations from the protocol. Therefore, it is ok to repeat the introduction and methods as this is simply the stage-2 of the same manuscript (using the same terminology as it is described in PlosOne guidelines). 

https://everyone.plos.org/2021/03/30/registered-reports-one-year-at-plos-one/

Indeed, some articles previously published as registered protocol and then as registered report research in the journal did not change the introduction and M&M when publishing the full results. 

Reviewer 1

This systematic review has investigated the measurement properties of pain scoring instruments intended for farm animals. The study reports in detail on the measurement properties of 20 tools for three species, cow, sheep and pig. The study also describes the nature and purpose of the properties. A concensus-based guideline made for selection of human health measurement instruments is used as a protocol.

This type of critical reviews of relatively new clinical methods are highly necessary and welcomed. The study is thorough and well planned. Despite that there are no original findings of the study, I find it very valuable for the future research. Not surprisingly, the study shows that some scales have better performance parameters than others. This probably mainly reflects the need of studies like this: a systematical approach to validation has not yet become a part of the pain scale developers mind sets.

Response: Thank you for your comments and taking the time to review the manuscript.

My main critical comments concerns the rather un-reflected use of COSMIN – is it really that easy to apply a human tool intended for verbal self-report, to a situation where humans observe animals for pain? I would like a commentary on that in the Introduction or Discussion, where it fits best. In relation to that, since there is no gold standard for pain in animals, do the authors have any suggestions for modification of the guideline. No scale is better than its construct validity, and the discussion becomes rather technical, where it would be nice to get an impression of the consequences of certain poor or missing properties, if possible.

Response: Thank you for the general comment. The first question can be subjective but our impression is that it is very reasonable to apply the COSMIN in such studies as previously reported by our group (Evangelista MC, Monteiro BP, Steagall P V. Measurement properties of grimace scales for pain assessment in non-human mammals: a systematic review. Pain. 2022 doi:10.1097/j.pain.0000000000002474), especially with adaptations to circumvent some limitations related to pain scoring instruments as described in the last paragraph of the discussion. Therefore, we have indicated where the guidelines required modifications. We have added a paragraph to the discussion to provide more information on this. We used the GRADE approach that has been suggested for systematic reviews of patient-reported outcomes to strengthen our methodology. As much as there are limitations with the COSMIN, to the authors’ knowledge, this is the only suitable method for this type of grading.

I below give a number of minor comments, some are merely typos.

Line 44: Most... Give number for clarity .

Response: Done

Line 47. Llama, Alpaca?

Response: Corrected to camelids

Line 57. ..Or low empathic capacity of farmers (barn blindness).

Response: Reworded

Line 66: Also, they do not necessarily measure the suffering component of pain.

Response: Added. 

Line 66. Other surrogate measures... lameness and activity are also surrogates?

Response: Yes.

Line 68... Are also not necessarily..

Response: Corrected.

Line 80: what is meant by appearance here?

Response: This is specifically described by Oliveira et al. 2017 (Validation of the UNESP-Botucatu unidimensional composite pain scale for assessing postoperative pain in cattle) without further details. It is presumed by the authors that it could be related to both physical and behavior aspects. 

Line 83: What is meant by curved lips in a cow?

Response: Replaced by “increased tonus of the lips” as described by Gleerup et al. 2015

Line 92: Why is species specificity of importance?

Response: Sorry maybe we did not understand the question. How would we be able to use a pain assessment tool that was validated for use in humans in cats, for example?

Line 138: peer-reviewed?

Response: Corrected. Thank you.

Line 187: this would be helpful as supplementary material.

Response: This information is all reported within the Tables and Supplementary material.

Line 197: Helpful as supplementary material.

Response: Same as above.

Line 199-204: This section belongs to Introduction, where nothing is mentioned about COSMIN. As it is in the title, a small introduction would be helpful.

Response: Thank you for your suggestion, but we respectfully prefer to keep the information about the COSMIN in the methods as we present what changes and modifications were performed to their guidelines. Even if the COSMIN is not presented in the introduction, it is still provided early in the methods.

Table 2, 2nd last section. What do Action Units mean in this context?

Response: Action units are described in grimace scales and they are the representation of individual components of muscle movements of the face. They are usually the components evaluated/scored during pain assessment and are first mentioned in the section ‘Data extraction’.

Table 4: Last row. How can the study be included if scoring method is not available?

Response: Scoring method herein is related to either video or real-time assessment and this information was not available for the Porcine BPSP, but it doesn’t mean that scoring was not performed. This was not a criterion for study exclusion. 

Line 299: there -instead of they?

Response: Thank you for picking this up. Corrected. 

Line 381. Content validity needs a more in depth discussion. As already mentioned, there is no gold standard for pain, and has consistently been shown that expert opinions differ much. Therefore, a more critical approach to content validity is warranted. Scales are no better than their content validity, despite excellent rater agreements and other criteria.

Response: We have expanded our discussion according to both reviewers’ suggestion. 

Line 389. You could add here that content validity of pain scales is a problem, since there is no gold standards to rely on. In addition, studies has shown considerable overlap between behaviours in pain and during stress, putting pressure on the correct identification of pain.

Response: Respectfully, the lack of gold-standard is related to criterion validity. This has been discussed on lines 438 and beyond. 

Line 399: Please explain the hypothesis behind why this is relevant. Is it the frequency of facial action units, which is meant? And why should that be correlated to the sum om AUs?

Response: Added. 

Line 401: Is it your opinion ICC is used correctly in all cited papers?

Response: To the best of our knowledge, the ICC was used correctly for inter- and or intra-reliability in the six manuscripts reporting ICC. Please state if the reviewer does not believe this is the case.

Line 407: So intra-rater agreement was always done on footage. Please explain how it should be done.

Response: Intra-rater reliability is performed using video or image assessment during validation of a pain scoring instrument. To the authors’ knowledge, this is the only way that this can be done. The comment is not clear. The discussion includes the issue related to short intervals applied to repeat video or image assessment during intra-rater reliability. 

Line 423: The global judgement is used in many pain publications in order to avoid circularity because of including the items of the scoring.

Response: Correct. However, it does not mean it should be applied on its own for the validation of a pain scoring system as it is highly subjective and may be biased, especially when not taking in consideration the scores.

Line 437. Using unidimensional scales does not make pain scoring better! The numbers looks good, but please explain why you think they can be used for criterion validity. If they were pain scales, we did not need to develop new scales.

Response: The authors never stated that unidimensional scales make pain scoring better or that they should be necessarily used, or that they are pain scales. We simply stated that, in the lack of other validated instruments or gold-standard, unidimensional scales have been or are used for initial criterion validity in many studies as they have been used in the past to evaluate the construct (i.e. pain). This is why none of the scales received “high” strength of evidence for this as comparators had poor validity. Indeed, this sentence criticizes the use of these unidimensional scales as they are not species-specific, for example. There is not a consensus on how criterion validity should be performed.

Line 441: Weak argument, since cortisol concentrations are not specific to pain, as you already have mentioned.

Response: Reworded to avoid confusion.

Line 453: Yes, this is actually a good discussion. As you already mention, there is no way to measure the degree of pain experience, are high pain scores sign of high pain intensity, or high pain probability or both? Disease: we have a biological understanding of pain pathology, and pathology (inflammation, trauma, etc.) may be a good proxy for pain probability, just as we accept for example lameness as a pain proxy.

Response: Agreed. 

Line 457: Are pain scales linear? Which statistics should be used to show significance or what is meant by significant?

Response: Pain scales are not usually linear. Responsiveness relates to the ability of an instrument to detect changes overtime in the construct to be measured (i.e. pain). Wilcoxon signed rank tests are normally used to compare the scores between, for example, before and after the administration of analgesics (item 2F on Table 2). The word “statistically” was added to the sentence.

Line 461. I believe that study 35 was based on analgesic testing, this study is not mentioned.

Response: Responsiveness was not assessed for study 35. Although the authors of that study report response to analgesic administration, this step was done during the development of the scale (Study I; scale with 15 items) and not in the actual scale (Study II; scale with 6 items). Therefore, responsiveness for the cow pain scale remains unknown. Further information was added to the discussion. 

Line 482. The rigidity of COSMIN. Yes. Could you discuss if any items should be omitted or modified for use in animal pain scale development?

Response: We have added to the discussion a comment about the gender of observers involved in the development and validation of the pain scoring instruments and the number of individuals involved in content validity, as this is not described in the COSMIN guidelines. We did not look at any other COSMIN items that should be omitted or modified. Indeed, we are using the same methodology for a new systematic review. We believe that we found a very strict and robust methodology with protocol registration according to the PRISMA and using COSMIN. The strength of evidence was performed using a modified GRADE. Limitations of COSMIN are present as with any other proposed assessment instrument and discussed in the manuscript. We published the protocol beforehand for transparency and better reporting (Tomacheuski et al. 2021). Our databases and search terms were used according to our librarian recommendations and exported using COVIDENCE. 

Line 486: Could you discuss the consequences of using scales with different deficiencies?

Response: A comment has been added.

Reviewer #2: 

Based on the COSMIN guidelines for assessing health measure instruments the present review seeks to provide evidence of reliability, validity and sensitivity of pain scoring instruments for farmed animals. The review identified 20 pain assessment protocols based on the initial search and following screening (inclusion and exclusion criteria). The steps of the COSMIN guidelines are followed nicely and result in a comprehensive overview of the 20 publications. The review adds important information to the sparse knowledge on validity within the growing field of pain scoring scales, as it identifies the most frequent shortcomings of publications on these instruments, provided helpful recommendations for future publications and further validation of pain scoring instruments.

The manuscript is well-written and easy to follow. However, there are some issues, I would like to address.

Firstly, I was a bit puzzled, that the complete introduction and the following parts all the way through to line 214 are exact copies of the previous report Tomacheuski et al. (2021)?

Response: Thank you for your comments and taking the time to review our systematic review. This has now been addressed with the editorial office and the academic editor. According to the journal’s guidelines, Registered Report Research Articles report the results of all planned analyses previously published in the journal (Tomacheuski et al. 2021) and, if relevant, detail and justify all deviations from the protocol. Therefore, it is ok to repeat the introduction and methods as this is simply the stage-2 of the same manuscript (using the same terminology as it is described in PlosOne guidelines). 

https://everyone.plos.org/2021/03/30/registered-reports-one-year-at-plos-one/

Indeed, some articles previously published as registered protocol and then as registered report research in the journal did not change the introduction and M&M when publishing the full results. 

The materials and methods state, that no language restrictions were imposed. Based on this decision I wonder how correct translations were to be ensured and how evidence from journals were considered in this review? Were manuscripts eligible if they were not peer-reviewed?

Response: As stated in the methods, the search only included peer-reviewed journals. Correct translations should not be a problem for the search and screening as articles in different languages usually have at least an abstract and key words in English. This was also not limited by the fact that five languages are fluently spoken within the group of authors. Additionally, we used strict and robust databases and search terms, eligibility criteria, literature search and data extraction, as suggested by our librarian. In terms of evidence, the quality assessment and summary of evidence were also strict using two independent reviewers with all the information recorded, evaluated and adapted from the COSMIN checklist. 

The definition of farm animals in lines 186-195 might be nice to have already before search terms are listed.

Response: Not sure about this comment as this entire paragraph describes how data extraction was performed, and not the definition of farm animals. The section ‘Eligibility Criteria’ includes description on what species were considered and why in this systematic review. We believe it makes more sense to have how data extraction was performed after eligibility criteria and literature search.

Regarding the exclusion criteria, why was the sample size of the included studies not considered? This could also affect validity of study outcomes.

Response: Our exclusion criteria included: Studies reporting the use of pain scoring instruments to measure constructs other than pain, for example studies assessing animal welfare, in which pain was considered within the overall evaluation, studies assessing nociceptive testing, and studies for which the full text was not available. Assessment of evidence was based on the COSMIN guidelines; therefore, the sample size is not specifically evaluated in this sense. However, the criteria for items 1a.3 and 1a.4 (Table 1) describe the target population and the sample representing this, which was taken in consideration. Indeed, detailed population characteristics for these studies are included in the supplementary material (Table S2).

The results are nicely illustrated in the two tables, however, the figure 1 has a really low resolution in the pdf-version distributed.

Response: All figures correspond to the journal’s guidelines so the authors are not sure what may have happened during the creation of the PDF.

Albeit the COSMIN approach ensures a good evaluation of the validity issues in regards to pain assessment protocols, the consequences of this rigid protocol are not really discussed in depth. Since most of the pain scoring instruments only are described by one publication (maximum four publications for the Porcine/PBS-B), there is actually not a lot of evidence yet. Hence, the discussion could be improved by adding a discussion of the included instruments' contents i.e. their feasibility and applicability and highlight their shortcomings in order to point out the remaining gaps of knowledge.

Response: Thank you for your suggestion. We agree that not a lot of evidence is available and this is one of the critical points of the systematic review. However, the manuscript is already long in length and we feel this has been addressed in the discussion. For example, the limitations of using the COSMIN are discussed in the second paragraph: “The majority of pain scoring instruments presented overall ‘low’ and ‘very low’ strength of evidence [14,17,24,35,50–56,59] due to a small number of studies available, inadequate methodological quality, and/or conflicting or indeterminate quality of findings according to the COSMIN guidelines” and “low ratings are potentially related to the rigorous of the COSMIN guidelines since the final score for each category is the lowest score from all criteria within that category. In other words, regardless of how many ‘very good’ or ‘moderate’ ratings a study received for different criteria, the rating would be ‘low’ if one of these criteria was scored as ‘low’”. More specifically, the last paragraph states “The small number of studies for most instruments or unclear reporting may have reduced the overall strength of evidence of measurement properties of pain scoring instruments”. Finally, the feasibility and interpretability of these instruments were evaluated during data extraction and in the results, we have reported that none of the studies have reported these features. We have now added a comment on this in the last paragraph.

Additionally, a discussion of the definition of pain would be beneficial, as it is a subjective sensory and emotional experience, one could argue that it also depends on the level of empathy of the observer.

Response: The aim of this systematic review was to provide evidence relating to the measurement properties (i.e. reliability, validity and sensitivity) of pain scoring instruments used for pain assessment in farm animals using the COSMIN. The issue of observer/gender variability is related to pain assessment itself and not the measurement properties of the scale and it is a complex issue that goes beyond the aims of our study. A paragraph has been added to the discussion to address this issue.

In case of expert opinion being used as to ensure content validity, the included studies ranking high on this measure did only consult 4 experts. How much evidence can four experts generate? How many experts would be required? I think this also needs to be addressed in the discussion.

Response: A comment has been added to the third paragraph of the discussion. According to the book ‘Streiner DL, Norman GR. Heath Measurement Scales: A practical guide to their development and use. 4th ed. Oxford, UK: Oxford University Press, 2008’, four to five experts are considered adequate for initial content validity of a health care instrument. We would like to highlight that content validity is also based on an index, development of ethograms and literature findings.

Finally, in the conclusion it would be nice, if the 'gaps of knowledge' were described again to emphasize what needs to be considered when planning the development and publishing of new pain assessment methods.

Response: Added

---

## [Decision Letter · Decision Letter 1]

13 Dec 2022

PONE-D-22-02200R1Measurement properties of pain scoring instruments in farm animals: a systematic review using the COSMIN checklistPLOS ONE

Dear Dr. Steagall,

Thank you for submitting your manuscript to PLOS ONE. After careful consideration, we feel that it has merit but does not fully meet PLOS ONE’s publication criteria as it currently stands. Therefore, we invite you to submit a revised version of the manuscript that addresses the points raised during the review process. Please submit your revised manuscript by Jan 27 2023 11:59PM. If you will need more time than this to complete your revisions, please reply to this message or contact the journal office at plosone@plos.org. Please include the following items when submitting your revised manuscript:A rebuttal letter that responds to each point raised by the academic editor and reviewer(s). You should upload this letter as a separate file labeled 'Response to Reviewers'.A marked-up copy of your manuscript that highlights changes made to the original version. You should upload this as a separate file labeled 'Revised Manuscript with Track Changes'.An unmarked version of your revised paper without tracked changes. You should upload this as a separate file labeled 'Manuscript'.If applicable, we recommend that you deposit your laboratory protocols in protocols.io to enhance the reproducibility of your results. Protocols.io assigns your protocol its own identifier (DOI) so that it can be cited independently in the future. For instructions see: https://journals.plos.org/plosone/s/submission-guidelines#loc-laboratory-protocols. Additionally, PLOS ONE offers an option for publishing peer-reviewed Lab Protocol articles, which describe protocols hosted on protocols.io. Read more information on sharing protocols at https://plos.org/protocols?utm_medium=editorial-email&utm_source=authorletters&utm_campaign=protocols.

We look forward to receiving your revised manuscript.

Kind regards,

Ali Montazeri

Academic Editor

PLOS ONE

Journal Requirements:

Reviewers' comments:

Reviewer's Responses to Questions

**Comments to the Author**

1. Does the manuscript adhere to the experimental procedures and analyses described in the Registered Report Protocol?

If the manuscript reports any deviations from the planned experimental procedures and analyses, those must be reasonable and adequately justified.

Reviewer #2: Yes

2. If the manuscript reports exploratory analyses or experimental procedures not outlined in the original Registered Report Protocol, are these reasonable, justified and methodologically sound?

A Registered Report may include valid exploratory analyses not previously outlined in the Registered Report Protocol, as long as they are described as such.

Reviewer #2: Yes

3. Are the conclusions supported by the data and do they address the research question presented in the Registered Report Protocol?

The manuscript must describe a technically sound piece of scientific research with data that supports the conclusions. The conclusions must be drawn appropriately based on the research question(s) outlined in the Registered Report Protocol and on the data presented.

Reviewer #2: Yes

4. Have the authors made all data underlying the findings in their manuscript fully available?

Reviewer #2: Yes

5. Is the manuscript presented in an intelligible fashion and written in standard English?

Reviewer #2: Yes

6. Review Comments to the Author

Please use the space provided to explain your answers to the questions above. (Please upload your review as an attachment if it exceeds 20,000 characters)

Reviewer #2: Thank you for the revised manuscript and clarifying answers in the attached response letter. My comments have been addressed more or less sufficiently apart from the issue of sample sizes.

Although, sample sizes are given in Table S2 and should be covered by your procedure according to Table 1, the issue of small vs. larger sample sizes does play a pivotal role for the specific validity measures of each study in terms of ICC/IOR and accuracy estimates. E.g. for cattle pain scoring instruments sample sizes in the included studies vary from as little as 8 to 345 animals. I think this is worth mentioning in the discussion.

Table 6 - Are footnotes for the letters a-j missing?

7. PLOS authors have the option to publish the peer review history of their article (what does this mean?). If published, this will include your full peer review and any attached files.

Reviewer #2: **Yes: **Nina Dam Otten

---

## [Author Response · Author response to Decision Letter 1]

28 Dec 2022

Reviewer #2: Thank you for the revised manuscript and clarifying answers in the attached response letter. My comments have been addressed more or less sufficiently apart from the issue of sample sizes.

Although, sample sizes are given in Table S2 and should be covered by your procedure according to Table 1, the issue of small vs. larger sample sizes does play a pivotal role for the specific validity measures of each study in terms of ICC/IOR and accuracy estimates. E.g. for cattle pain scoring instruments sample sizes in the included studies vary from as little as 8 to 345 animals. I think this is worth mentioning in the discussion.

Answer: Thank you for reviewing the manuscript once again. A comment has been added to the discussion: “Additionally, results of reliability testing and other measurement properties could have been influenced by the sample size (i.e. number of animals included) among studies. Indeed, the COSMIN criteria do not take study sample size in consideration during methodological quality assessment”.

Table 6 - Are footnotes for the letters a-j missing?

Answer: Footnotes have been added.

---

## [Editor Report · Decision Letter 2]

10 Jan 2023

Measurement properties of pain scoring instruments in farm animals: a systematic review using the COSMIN checklist

PONE-D-22-02200R2

Dear Dr. Steagall,

We’re pleased to inform you that your manuscript has been judged scientifically suitable for publication and will be formally accepted for publication once it meets all outstanding technical requirements.

Kind regards,

Ali Montazeri

Academic Editor

PLOS ONE

---

## [Editor Report · Acceptance letter]

12 Jan 2023

PONE-D-22-02200R2 

Measurement properties of pain scoring instruments in farm animals: a systematic review using the COSMIN checklist 

Dear Dr. Steagall:

I'm pleased to inform you that your manuscript has been deemed suitable for publication in PLOS ONE. Congratulations! Your manuscript is now with our production department. 

Kind regards, 

on behalf of

Professor Ali Montazeri 

Academic Editor

PLOS ONE